# The Effects of Handedness Consistency on the Identification of Own- and Cross-Race Faces

**DOI:** 10.3390/brainsci15080828

**Published:** 2025-07-31

**Authors:** Raymond P. Voss, Ryan Corser, Stephen Prunier, John D. Jasper

**Affiliations:** 1Department of Psychology, Purdue University Fort Wayne, Fort Wayne, IN 46805, USA; 2Owen Graduate School of Management, Vanderbilt University, Nashville, TN 37203, USA; ryan.j.corser@vanderbilt.edu; 3Department of Psychology, University of Toledo, Toledo, OH 43606, USA; john.jasper@utoledo.edu; 4Department of Psychology and Vice Chancellor for Academic Affairs, Ivy Tech Community College, Indianapolis, IN 46208, USA; sprunier@ivytech.edu

**Keywords:** handedness, own-race bias, laterality, hemispheric specialization, cross-race effect, facial recognition

## Abstract

**Background/Objectives**: People are better at recognizing the faces of racial in-group members than out-group members. This own-race bias relies on pattern recognition and memory processes, which rely on hemispheric specialization. We hypothesized that handedness, a proxy for hemispheric specialization, would moderate own-race bias. Specifically, consistently handed individuals perform better on tasks that require the hemispheres to work independently, while inconsistently handed individuals perform better on tasks that require integration. This led to the hypothesis that inconsistently handed individuals would show less own-race bias, driven by an increase in accuracy. **Methods**: 281 participants completed the study in exchange for course credit. Of those, the sample was isolated to Caucasian (174) and African American individuals (41). Participants were shown two target faces (one Caucasian and one African American), given several distractor tasks, and then asked to identify the target faces during two sequential line-ups, each terminating when participants made an identification judgment. **Results**: Continuous handedness score and the match between participant race and target face race were entered into a binary logistic regression predicting correct/incorrect identifications. The overall model was statistically significant, *Χ*^2^ (3, *N* = 430) = 11.036, *p* = 0.012, *Nagelkerke R*^2^ = 0.038, culminating in 76% correct classifications. Analyses of the parameter estimates showed that the racial match, *b* = 0.53, *SE* = 0.23, *Wald Χ*^2^ (1) = 5.217, *p* = 0.022, OR = 1.703 and the interaction between handedness and the racial match, *b* = 0.51, *SE* = 0.23, *Wald test* = 4.813, *p* = 0.028, OR = 1.671 significantly contributed to the model. The model indicated that the probability of identification was similar for own- or cross-race targets amongst inconsistently handed individuals. Consistently handed individuals, by contrast, showed an increase in accuracy for the own-race target and a decrease in accuracy for cross-race targets. **Conclusions**: Results partially supported the hypotheses. Inconsistently handed individuals did show less own-race bias. This finding, however, seemed to be driven by differences in accuracy amongst consistently handed individuals rather than a hypothesized increase in accuracy amongst inconsistently handed individuals. Underlying hemispheric specialization, as measured by proxy with handedness, may impact the own-race bias in facial recognition. Future research is required to investigate the mechanisms, however, as the directional differences were different than hypothesized.

## 1. Introduction

Humans are often described as social animals. Although this description might overgeneralize the complexity of human beings, it does place accurate emphasis on the social nature of our species. Because of this social nature, humans seem to be tuned to certain aspects of personal interactions. Chief among these is the ability to process and recognize faces, which some argue is a distinct process separate from other forms of object recognition [1]. These abilities have real-world impact. For example, they have been purported to affect facial recognition processes involved in eyewitness testimony [2] and the identification of cross-race subjects [3].

With this in mind, research into facial recognition processes has increased dramatically in recent years, becoming one of the most studied topics in cognitive psychology [4]. Previous research has found evidence for both a potential unique mechanism of facial recognition [5] as well as potential evidence of a learned expertise [6]. In support of the former, several studies have found evidence for specific neural correlates of facial recognition, most famously in the occipitotemporal regions (such as the fusiform face area [7]) and increased activation in the right hemisphere [8,9]. Research regarding the latter indicates that facial recognitions may at least partly rely on past experiences and learning [10]. However, the effect of experience may differ across participants as a result of facial recognition skill [11]. Interestingly, both of these potential mechanisms could share ties with another individual difference measure, namely an individual’s handedness.

A growing body of research indicates that consistency of handedness impacts memory, right-hemisphere-based processing, and use of stereotypic information [12,13]. Specifically, inconsistently handed individuals seem to have better episodic memory for both word lists and actual events [14,15]. Additionally, their advantage with right-hemisphere-based processes and the integration of the cerebral hemispheres [16] could allow inconsistently handed individuals greater access to processes that have been shown to impact facial recognition; such as holistic processing [17], coordinate-based processing [18], or even direct processing of cross-race targets [19].

In addition to recognition abilities, other evidence suggests that consistency of handedness also influences other aspects of facial processing, such as emotions [20] and ratings of attractiveness [21]. Beyond facial processing, research also indicates that consistency of handedness is related to stereotyping and prejudice, with inconsistently handed individuals being more likely to remember stereotype inconsistent information [22] or more likely to be affected by false feedback accusing them of prejudice [23]. Taken together, it is reasonable to predict that consistency of handedness may moderate the own-race bias. Despite this, no known studies have investigated their effects together. In this paper, we aim to investigate how handedness may affect the own-race bias often observed with facial recognition tasks.

### 1.1. Own-Race Bias

The own-race bias in facial recognition (also known as the cross-race effect or the other-race effect) is a bias in which people tend to be more accurate identifying and remembering the faces of other individuals when they (the judges) belong to the same racial in-group of those being judged [24]. Recent meta-analyses seem to indicate that the bias is reliable, though small [3,25,26,27,28]. Although the effects are small, an increased understanding of the variables that influence cross-race identifications could be important for ensuring both social justice and an equitable legal system in America, especially given the recent difficulties in race relations in the U.S., highlighted by the challenges the country faced during 2020 and 2025. In fact, the own-race bias has been implicated by the Innocence Project in at least 40% of the DNA exonerations where individuals were convicted based primarily on eyewitness testimony.

Many researchers attribute own-race bias to a practice effect, caused by exposure and interaction with an individual’s own race. One piece of supporting evidence shows that people become experts at processing facial features of others that the individual interacts with regularly. Highlighting this, early research found that black and white participants self-reported a collection of different features when distinguishing faces, many of which would offer little identifying information for a cross-race target [29]. For example, white participants were more likely to identify iris color, hair color, and hair texture which are more distinguishing features for white targets. In contrast, black participants were more likely to use global facial descriptions, and to mention hair position, eye-size, whites of the eye, eyebrows, ears, and chin—features that tend to be more salient for black targets. Offering further evidence for an experiential basis to the effect, Sangrigoli and colleagues [30], examined the own-race bias in Korean adults, some of whom were adopted by Caucasian families. They found that the adopted individuals were better at recognizing white faces than Asian faces whereas the opposite was true for the control group of Korean adults. In fact, previous research has indicated that experience with races other than one’s own seems to attenuate the own-race bias [28]. Several potential mechanisms have been proposed to account for this experience effect, chiefly perceptual learning, wherein an individual’s exposure to other faces, especially in early childhood, allows for a tuning of the expertise system to faces that are most commonly viewed [31]. Recent research, however, has produced challenges to this expertise account [32]. Whatever the true mechanism, meta-analyses still indicate that experience accounts for a small, but reliable, amount of variance across studies [26,27,28].

Despite its popularity, the experience account is not the only theoretical explanation for the observations of the own-race bias. Other researchers have proposed that the differences seen are the result of differences between processing styles used for own- vs. cross-race targets. In one theory, researchers argue that own-races faces are more likely to be processed using configural processing, while cross-race faces are more likely to be processed using featural processing (see [33] for an example). In a similar vein, another theory proposed by Levin [34] focuses on different cognitive processing as the mechanism for the ORB. Specifically, this theory argues that people are likely to make a categorical judgment about cross-race targets and focus on the features that classify them as a different race. When viewing targets of the same race, however, people are likely to individuate and process the features of the target with enough detail to aid in later recognition. These theoretical accounts all tie directly to processes linked to differential hemispheric activation and handedness. Specifically, research indicates that inconsistently handed individuals have better episodic memory (experience account), are more likely to rely on right-hemisphere-based processing (processing style account), and are less likely to stereotype outgroups, potentially as the result of their cognitive categorization (cognitive account) [12,13].

It is important to recognize that additional theories for the ORB exist beyond those directly discussed above, such as the virtual face space model [35], the in-group/out-group model [36], the categorization–individuation model [37], and the perceptual social-linkage theory [38], to name a few. Though the exact mechanisms behind the own-race bias in facial recognition are still heavily debated, it is interesting that several of the proposed accounts point to mechanisms with known hemispheric asymmetries and links to handedness.

### 1.2. Hemispheric Specialization of Facial Recognition

Identifying the underlying neural processes of facial recognition has been a key concern for cognitive neuroscientists since the observation that brain damage could produce deficits in facial recognition without necessarily impairing other forms of object recognition (for an example, see Ellis & Florence’s translation of Bodamer’s classic paper on prosopagnosia) [39,40]. Later fMRI work showed that the fusiform gyrus, a region of the temporal lobe, was activated when individuals viewed faces [41]. Though this activation may not be unique to faces, as other work has found evidence that the fusiform gyrus is also activated when viewing members from a category that the individual has some level of expertise with [6], it is still an important indication concerning the underlying mechanisms, and importance, of facial recognition processes. More recent fMRI research has also shown a different pattern of activation between the right and left fusiform face areas. The left hemisphere (LH) fusiform face area seems to be activated when viewing both facial stimuli and stimuli that resemble faces. The right hemisphere (RH) fusiform face area, in contrast, shows greater activation for only facial stimuli [8]. Meng and colleagues used this to argue that there is a RH advantage for categorizing objects into a face or non-face.

Several other research streams show RH advantages for cognitive processes that may also be important for facial recognition. For example, facial recognition relies greatly on holistic (or gestalt)-based processing [42,43]. Facial recognition can be considered holistic because human beings tend to process faces as a whole and show impairment when recognizing individual facial components out of context [17,44]. This holistic perception even extends to full body representations wherein the face and body are integrated into a single perception, especially when processing emotional content [45,46,47,48]. In fact, this integration could be an extension of the RHs penchant for processing the contextual information of a situation [49,50]. This holistic processing is interesting because previous research has indicated that the RH has an advantage processing information holistically [17,51,52,53,54]. Similarly, research also indicates that facial recognition may rely on coordinate-based processing of the facial components [18], which again shows a RH advantage [51,55]. These advantages for holistic and coordinate based processing should lead to improved processing of faces within the RH.

Regardless of the exact mechanism, past research does in fact highlight the importance of, and perhaps advantage for, the RH in facial recognition processing. For example, the RH shows increased activation of face-specific neural areas (such as the fusiform face area, the occipital face area, and the posterior superior temporal sulcus), faster processing of faces displayed in the left visual field compared to the right visual field, the recognition of left-chimeric faces as the original over right-chimeric faces, and even the conscious experience of faces with direct electrical stimulation to the RH as compared to the LH face areas [56]. Additionally, and similar to the effects of consistency of handedness mentioned briefly above, there is also research support for hemispheric asymmetries in the processing of facial emotions [57] and attractiveness [58]. Taken together, this research seems to indicate that the RH is uniquely related to processing faces. Other research has also indicated that interhemispheric interaction is important for facial recognition processes [59].

### 1.3. Potential Relationship Between Handedness, Laterality, and the Own-Race Bias

To tie it all together, a recently published paper by Jasper and colleagues [16] provides a framework for investigating handedness as an important component of facial recognition. Specifically, inconsistently handed (ICH) individuals seem to have greater access to RH-based processes and perform better on tasks that require the integration of the cerebral hemispheres and dual processes, while consistently handed (CH) individuals seem to have an advantage with tasks that rely on independent processes. It is important to distinguish that ICH is not the same as ambidextrous. Where a true ambidextrous individual may be close to equal use of both hands across tasks, the inconsistent category is much broader and often features individuals who do have a hand preference but perform at least some tasks with their non-dominant hand. CH individuals, by contrast, perform almost all tasks with their dominant hand [12,13]. Research has implicated corpus collosum size as a probable structural and functional difference separating these groups, with inconsistent handers, on average, having larger corpus collosa [60] (for a thorough review of degree of handedness and lateralized cognition, see [12,13]). This increased corpus collosum size and interhemispheric communication should lead to greater use of these right-hemisphere-based processes as research indicates that communication from the right to the left hemisphere is quicker and contains more axonal projections [61]. This handedness difference could manifest itself with ICH individuals showing increased access to the previously discussed RH-based holistic and coordinate processing and the inherent advantages those bring for facial recognition, as well as the use of more integrated dual-processes, as outlined in the above theory. It is important to recognize that other competing theories of handedness and its relation to information processing also exist. Specifically, some theorists argue that direction is an important distinction for behavior rather than the consistency of handedness. Specifically, these theories often indicate that left handedness is related to reduced brain lateralization [62] and greater access to the RH and RH-based processes, such as visuospatial-based processing and memory recall [63]. (For additional reviews, see [64,65,66].) This specific study and research question, however, were formulated around the theory put forth by Jasper and colleagues and the impact of consistency of handedness which will be discussed further below.

Beyond just its associations with holistic-based processes in the RH, the degree of handedness influences other processes that may also have a direct effect on an individual’s ability to recognize cross-race faces. First, research indicates that ICH individuals have superior episodic memory [14]. These advantages hold for both word list memory as well as recall of actual events [15]. Specifically, the LH seems to play a key role in the encoding of episodic memories, while the RH plays a key role in the retrieval of episodic memories [67]. This purported superiority in the ability to recall episodic memories for ICH individuals in general could translate to an increased memory for the presented target face. Other research also seems to indicate that the cross-race effect is largely based on the encoding process [27] which could differentially influence these handedness groups.

Second, the consistency of handedness seems to be related to the application of prejudiced and stereotypic beliefs, with ICH individuals often showing fewer stereotyping behaviors compared to CH individuals. For example, during a memory task, ICH individuals showed greater memory for information that was inconsistent with commonly held stereotypes [22]. In a related study examining sexism and the impact of cognitive dissonance, ICH individuals were more motivated by randomly assigned feedback insinuating they were sexist and responded to future tasks in a manner consistent with overtly anti-sexist behaviors. Specifically, they awarded larger sums of money to a plaintiff in a fictional sexual harassment lawsuit [23]. This lessened use of common stereotypes for ICH individuals could lead to a reduction in the own-race bias as they may be more likely to view all faces equally, regardless of in-group/out-group membership.

More importantly, the RH has been directly implicated in the processing of cross-race faces. In one study, Prete and Tommasi [19] presented participants with own- and cross-race faces to the left visual field (RH), the right visual field (LH), or both visual fields (central presentation). Participants were instructed to categorize the presented faces into the correct racial category. During central presentations, the typical own-race bias emerged, with participants showing more accurate categorizations for own-race faces. Perhaps most importantly for the research presented in this report, an interesting interaction emerged during divided visual field presentations. Specifically, while investigating the Inverse Efficiency Score (the response times on correct decisions divided by the proportion of correct choices, smaller numbers indicate better processing), cross-race faces were processed better when presented in the left visual field (RH) than in the right visual field (LH), while own-race faces were processed better by in the right visual field (LH).

In summary, there is good reason to believe that the own-race bias in facial recognition could be influenced by participant handedness. Specifically, ICH individuals seem to rely more on the use of integrated dual-processes, processes localized to the right hemisphere, and on processes that require interhemispheric interaction. Taken together, all of these point to improvements in performance for ICH. Specifically, it was hypothesized that ICH individuals would show a decreased own-race bias as compared to CH individuals. That is, we predicted that ICH individuals would show less difference in the number of correct identifications between their own and different (other) races. We also hypothesized that this difference would be driven by an increase in recognitions, and an overall performance advantage, for ICH individuals.

## 2. Materials and Methods

### 2.1. Setting

Participants completed the study at a midsized midwestern university in a laboratory within the Department of Psychology. The laboratory was situated on the 5th floor surrounded predominantly by office space, and the doors were closed to minimize outside distractions. The room was lit by overhead fluorescent lighting. Participants completed the study at individual computer workstations that were separated by dividers and were accompanied by up to 3 other participants (and the experimenter) in the room at the same time. Participants completed the study across a single 30 min research session. Stimuli were presented using Empirisoft (New York, NY, USA) Direct RT™ v2014 which was embedded within a larger Medialab™ v2014 presentation.

### 2.2. Materials

Facial stimuli were created by using mugshot-style photographs of inmates obtained from a publicly available state department of corrections website. All photographs featured female inmates who were filtered to have similar ages, weights, heights, and physical descriptions. As previous research has also indicated facial processing differences based on the gender of a target face [68,69], the study was isolated to female faces in order to eliminate any potential interactions with the cross-race effect. Faces from the database that featured scars, tattoos, or other easily identifiable characteristics were discarded. In total, 24 photographs featured white/Caucasian individuals and 24 featured black/African American individuals. Images were placed on a black background, edited to appear grayscale and masked with a black overlay to isolate only the interior facial features of the individuals. As the photos were all selected based on similar physical characteristics in the system, they were then randomly split to form two sets of white/Caucasian faces (with 12 photos in each set) and two sets of black/African American faces (again with 12 faces per set). A target face was then randomly chosen from each of the 4 sets. See Figure 1 for an example of the stimuli.

The overall size of the faces varied as the individual’s natural features were used to create the overlay. All images were placed on a black background and created with overall brightness levels ranging from 15 to 36.58 in mean pixel intensity. Overall contrast was measured as the standard deviation of pixel intensity and ranged from 27.99 to 71.07. Higher values corresponded to white faces. The computers in the laboratory were set to 50% brightness, but participants were not monitored for any adjustments to the computer display settings.

Participant handedness was measured using a slightly modified version of the Edinburgh handedness inventory [70] commonly used in handedness research [12,13,71]. Specifically, participants rated their use of hands on 10 common everyday manual tasks (writing, drawing, throwing a ball, using a spoon, using a knife, opening a jar, using a toothbrush, using a comb, using scissors, and lighting a match) on a 5-point Likert-type scale ranging from “always left” to “always right”. Finally, participants completed a standardized laboratory demographic questionnaire where they self-identified items such as their age, race, gender, year in school, grade point average, and whether they lived on campus. Participants also indicated how many siblings they had and whether any immediate family members were left-handed.

### 2.3. Participants

#### 2.3.1. Overall Sample

A total of 281 participants completed the study. Participants were recruited from introductory psychology courses and completed the study in exchange for course credit. Participants were given a list of potential studies in an online database and self-selected their research study from the list. Participants were given the opportunity to complete an alternate assignment if they did not wish to participate in the research. The data for 1 participant was lost due to a coding error, leaving 280 participants available for analysis.

#### 2.3.2. Racial Distribution

Of the available 280 participants, 174 participants self-identified as white or Caucasian and 41 self-identified as black or African American. These participants represented the target populations of the study and were analyzed further. The remaining participants either identified as another race (Middle Eastern/Arabic = 19, Asian = 13, Hispanic/Latino = 11), were of mixed races (13), or did not disclose (9); these participants were not analyzed further.

#### 2.3.3. Biological Sex Distribution

Of the participants who self-identified as white or Caucasian, 94 self-identified as female (M_age_ = 19.02, SD = 3.18) and 80 self-identified as male (M_age_ = 19.44, SD = 1.77). Among the black/African American participants, 29 self-identified as female (M_age_ = 18.96, SD = 1.38) and 12 self-identified as male (M_age_ = 19.25, SD = 1.66).

#### 2.3.4. Handedness Distribution

Following previous research [12,13,71], handedness was scored according to the following scale (always left = −10, usually left = −5, no preference = 0, usually right = +5, always right = +10). The scores were then added up and the absolute value was taken. Thus, lower values on the scale correspond to more inconsistent handedness while higher scores correspond to more consistent handedness. As a descriptive measure of the sample, a median split was performed and indicated that the sample comprised 101 inconsistently handed individuals (EHI scores between 0 and 70) and 114 consistently handed individuals (EHI between 75 and 100). Of the consistently handed individuals, 10 were consistently left-handed and 104 were consistently right-handed. See Table 1 for a complete breakdown of handedness frequencies.

### 2.4. Procedure

Participants entered the lab and were seated at one of the individual workstations. They were then provided with an informed consent form and given the time to read the form and ask any questions they had about the study or the informed consent process. After participants provided consent, the laboratory door was shut to minimize external noise from the hallway. Participants were then instructed to begin the study and follow the directions presented on the computer screen.

Participants were then randomly assigned to conditions and shown two target faces (1 white and 1 black), one at a time, each for 3 s. The order of the target faces was counterbalanced across all conditions. The stimuli set that the target face was chosen from was also counterbalanced across conditions. Participants were instructed that they would need to identify the target faces they saw at some later point in the study. Participants were given only a single target face from each racial group in an attempt to increase external validity by making the task similar to a real-life situation where they would be asked to identify a single individual or a small group of people. This also limited the overall number of recollections the participants had to complete. Previous research has indicated that having participants perform similar tasks repeatedly can alter their processing of the task, seemingly by moving the task toward more automatic processes [72]. The main focus of the study was the potential application of this work. Limiting decisions to only a target face of each race should increase the likelihood that participants’ behavior matches what could be expected outside of a lab setting.

Following this presentation, participants completed a series of distractor tasks that included a common numeracy scale, a ratio-bias task, a framing task, and a product evaluation task. These tasks were intended to occupy the participants so they could not actively focus on the target faces. Participants were free to complete the tasks at their own pace. These tasks served only as distractions and will not be discussed further.

Interspersed with the distractor tasks was the demographic questionnaire and the modified Edinburgh handedness inventory (12–13, 72–73).

Following this material, participants were presented with a series of faces and instructed to indicate if they had seen the face before. Faces were presented using a sequential photograph line-up as suggested by some of the eye-witness literature [2,73,74]. It is also important to recognize though, that other research disagrees with this recommendation [75]. Despite these inconsistencies within the literature, and the fact that the national research council (as discussed in [76]) and a recent scientific review of policy recommendation [77] left off recommendations for sequential procedures, some jurisdictions had already incorporated the research into their own policies [78], so this method was maintained to further accentuate the applied nature of the study. Participants were instructed that the target face may or may not be present in the line-up. Face fillers comprised the remaining faces in the chosen sets. The fillers and the target face within each set were presented to participants in a random order. Specifically, participants saw a single face at a time and were asked to make a Yes/No judgment (Do you recognize this face from earlier in the experiment?). Participants were allowed to make these judgments at their own pace.

Consistent with the sequential line-up discussed by Pickel [2], once participants made a “Yes” judgment (whether correct or incorrect) that portion of the study ended. If participants viewed all faces in the set without making a “Yes” judgment, that portion of the study also ended. Though the termination of the study upon a yes judgment may align with some practices in certain jurisdictions, it is not endorsed by all researchers. While the intention was to again align with potential real-world applications, the procedure does limit potential analyses. Participants viewed a single face set (corresponding to one of the two target faces) at a time. Each set contained a maximum of 12 faces, and the presentation order of the line-ups was counterbalanced across conditions.

Following the two identification tasks (one with white and one with black faces), participants were debriefed about the true purpose of the study, encouraged to ask questions, and were then dismissed.

## 3. Results

Prior to analysis, the absolute value handedness variable was z-score standardized to center the variable. This continuous handedness variable was then entered as a predictor in the analyses. As each participant saw a target face from each racial group, the participant race and race of the target face were coded together as a single variable. This variable represented either a match between races or a mismatch. This variable was then used as a within subjects’ factor in the analyses.

Because each participant saw only one target face for each racial group, and the task ceased when a participant identified a face (whether the identified face was correct or not) or when they exhausted all the options, the dependent variable of the task was essentially a binary response variable (between correct identification of the target and incorrect identification/missed identification) within each racial group lineup. To examine this binary choice, the data were analyzed using a logistic regression within IBM (Armonk, NY, USA) SPSS v29 with follow-up analyses performed using PROCESS v4.3.1 [79].

The overall model was statistically significant, *Χ*^2^ (3, *N* = 430) = 11.036, *p* = 0.012, *Nagelkerke R*^2^ = 0.038, culminating in 76% correct classifications. Analyses of the parameter estimates showed that the match variable, *b* = 0.53, *SE* = 0.23, *Wald Χ*^2^ (1) = 5.217, *p* = 0.022, OR = 1.703 and the interaction between handedness and the match variable, *b* = 0.51, *SE* = 0.23, *Wald test* = 4.813, *p* = 0.028, OR = 1.671 significantly contributed to the model (see Table 2 for the full parameter estimates of the model).

Visual inspection of the logistic model was conducted using sheets created by Dawson, see Figure 2 [80]. This visual inspection indicated that inconsistent handedness results in less own-race bias, with predicted probabilities being similar whether the race of the target face matches the participant’s race or not. As you move further toward consistent handedness, however, the bias seems to increase, with consistent handedness resulting in increased accuracy for same race targets and lower accuracy for mismatched targets.

To follow-up on these analyses, the conditional effect of the Match variable was tested at various levels of handedness using PROCESS [79]. No effect of match emerged when tested at 1 standard deviation below average for handedness, *b* = 0.019, *SE* = 0.331, *Z* = 0.056, *p* = 0.955. When tested at the handedness average, *b* = 0.532, *SE* = 0.233, *Z* = 2.284, *p* = 0.022, and one standard deviation above average, *b* = 1.046, *SE* = 0.330, *Z* = 3.172, *p* = 0.002, however, the effect of match was significant. Specifically, this indicates that the cross-race effect is non-significant amongst inconsistently handed individuals measured at −1 SD below the mean, and increases as participants become more consistently handed. The Johnson–Neyman significance region was identified as −0.138 with 39.535% of scores below this point, indicating that the effect of match was statistically significant for handedness scores beyond this point.

As an additional follow-up, the descriptive type of decisions made in each condition were also investigated. Specifically, when there was a racial match between the participant and the target face, 62 participants correctly identified the target, 146 selected an incorrect filler face before ever seeing the target face, 1 selected an incorrect filler after having missed the target, and 6 individuals did not select any face. While looking at the cross-race conditions, 42 participants correctly identified the target, 166 selected an incorrect filler before ever seeing the target face, 2 selected an incorrect filler after having missed the target face, and 5 individuals did not select a face. See Table 3.

## 4. Discussion

Based on previous research, we hypothesized that the race of the target face, the race of the participant, and participant’s handedness would be related to accuracy on a facial recognition task. More specifically, we expected ICH individuals to show an attenuated own-race bias compared to CH ones. It was also speculated that this difference would be largely driven by increased accuracy in facial recognition amongst ICH participants.

Overall, the hypotheses were partially supported. Specifically, individuals with ICH showed less own-race bias than those with CH. Interestingly, as evidenced in Figure 2, this difference was not driven by ICH, showing overall better recognition performance but instead was driven by CH, showing an increased accuracy for their own race and decreased accuracy for cross-race targets. ICH performance, however, was descriptively in the middle regardless of the target race.

It is this deviation from the hypothesized mechanism, however, that could provide a significant contribution to future research. Specifically, it seems that inconsistent hander’s episodic memory advantage [12,13] does not generalize to the facial recognition task in this study, suggesting that some other mechanism (besides memory processes) could explain the interaction. Several potential additional mechanisms exist that may explain the presented effects. First, research on perspective taking has pointed to potential differences between handedness groups [81,82], with ICH showing an increased ability to take other’s perspectives and an increase in memory for counter-stereotypic information [22]. It could be that ICH individuals are more likely to encode everyone as part of the same in-group (perhaps the human race), while those CH individuals are more likely to encode their own race as the in-group and other races as the out-group, leading to differences in processing same- vs. cross-race targets. In fact, research indicates that polarized groupings can lead to processing differences in faces of in- vs. out-group members (though this does not extend to all categorizations) [83]. Future research should explore the categorization of individuals amongst handedness groups to see if differences emerge in categorization strategies. This could also relate to the relationship between inconsistent handedness and context. Specifically, inconstant handers seem to be more influenced by the context of a choice [12,13,16]. It could be that this improved sensitivity to context results in a greater embeddedness in the world and allows inconsistent handers to process the cross-race targets more holistically and increase their likelihood of being incorporated into their in-group.

Another potential mechanism could lie in the type of processing utilized when recognizing cross- vs. same-race targets. Specifically, the own-race bias is often theorized as a decrease in accuracy for cross-race faces. In this specific study, however, it seems likely that the cross-race effect could be driven by an increase in performance when viewing same race targets amongst consistent handers. Based on the theoretical framework of Jasper and colleagues [16] this would seem to imply that the own-race bias in facial recognition may be the result of an independent process. Specifically, tasks that require an individual to select an egocentric perspective, which by definition forces you to ignore the perspectives of others, are more aptly handled by consistently handed individuals as they can independently process this egocentric frame. This is of particular interest when discussing racial differences because egocentrism is often discussed as a potential mechanism for general in-group biases [84,85]. It is possible that this egocentrism also contributes to the recognition of cross- and own-race faces. Future research should investigate the links between egocentric thought and the cross-race effect by having participants complete an egocentrism scale followed by completing a facial recognition task. Links here would imply it as a potential mechanism between handedness and the cross-race effect.

Although the specific mechanism is still undetermined, the evidence that handedness does impact the own-race bias in facial recognition provides additional theoretical contributions to our understanding of handedness. Specifically, handedness influences cognitive processing beyond the well-documented effects on memory and decision-making.

### 4.1. Practical Implications

Beyond the potential theoretical expansion of handedness and/or facial recognition research, the results reported herein may have practical implications for eye-witness testimony and suspect identification as well, especially within a line-up. If future research provides additional evidence for a systematic effect of handedness, a simple handedness survey (which typically takes less than a minute to complete) administered prior to a line-up could prove useful. This could be a double-edged sword, however, as an attenuation of the own-race bias seems to come at a reduction in accuracy for same-race targets. Future research is required to determine the effectiveness of such a course within an actual eye-witness context before any firm recommendations can be made. This is especially warranted because it is easy to imagine situations where a lawyer may argue that an eye-witness’s testimony is inaccurate simply because of their race or handedness, which may not be true. Perhaps, instead, the results of the survey could be used to aid in investigations when multiple eyewitnesses are present, but police resources are limited. Specifically, the handedness of the witness and their description of the perpetrator could be used to determine what leads to investigate first. Either way, additional investigations must first replicate the effects reported here before any of these uses can be applied to real-world cases.

Following this, the theory of Jasper and colleagues [16] would also imply that the own-race bias could be attenuated by encouraging an integration of the two hemispheres and their respective processes. Future research should investigate potential active manipulations for hemispheric activation, such as specific frequency presentation [86], saccadic eye movements [87], Schiffer goggles [88], and hand squeezing [89], to determine if differential hemispheric activation could improve performance on facial recognition tasks involving the cross-race effect. Specifically, increasing right hemispheric activation amongst consistently handed individuals should increase integrated processing and lead to a reduction in the bias. If this research supports an improvement in performance, then specific hemispheric presentation/activation could be utilized to achieve the most accurate testimony given the specific circumstances of the situation and the witnesses’ handedness.

### 4.2. Limitations

It is important to address the potential limitations of the current study design. One limitation is that participants viewed the same picture of the target during the learning and testing faces. Potentially, the results of the reported interactions could be due to specific image recognition, rather than being generalizable to facial recognition and the own-race bias. Specifically, participants could have relied on the overall shape of the faces, color of the skin, image brightness, or contrast to make their recognition judgments. If the effect were to be isolated to specific image recognition, or the other potential cues in the image, then the use of handedness for facial recognition or eye-witness testimony would be limited. If this were the case, further research into the cause of such a handedness difference could also be of interest, especially to determine what aspects of the image drive the recognition differences between the handedness groups. There are several reasons to think that the recognition of the specific image was not the cause of the reported effects, however. First, the majority of participants made an incorrect judgment before ever seeing the target image, and no filler image was selected significantly more than another. If they were relying on some form of more basic image recognition, then it must be a component that was easily confused across images. Second, an increase in overall memory amongst inconsistently handed individuals was not observed. With the documented increase in episodic memory for ICH, it would make sense for them to show a similar performance increase if the effects were relying on simple recognition of the previously seen target images. Future research is needed to establish the mechanisms underlying the observed effects in this study. Specifically, a study should be conducted using multiple pictures of the same faces to determine if the effects are generalizable to facial recognition overall or if the results are restricted to this specific task. Additionally, an investigation using full color faces would also help to elaborate on the reported effects. This study intentionally used grayscale images to reduce the potential impacts of skin color variance within each racial group. In actuality, this may be an important feature in facial recognition outside of the laboratory setting.

Another limitation of the present study is that it did not use signal-detection theory to investigate recognition performance. Specifically, by using the sequential line-up with a termination on any yes judgment, the study better aligned with the processes in place by certain law enforcement jurisdictions, but only investigated the impact of handedness on correct choices in the facial recognition. This process did not allow for investigations into the impact of handedness in relation to misses or false alarms. Before the results of this study can be generalized to all aspects of the own-race bias, it is imperative to investigate the effect using signal detection theory to determine if handedness would relate differentially to hits, misses, and false alarms. It is possible that the increase in accuracy for own-race targets in consistent handers may come at the price of increased false alarms or other detrimental outcomes. Similarly, the reduction in the bias in inconsistent handers may also be at some expense. In the present study, we presented only a single target from each race in an attempt to maintain external validity and examine how handedness may interact with the own-race bias during actual sequential lineups. Future work should investigate this with several iterations of own- and cross-race identifications before any recommendations can be made. Additionally, the current methodology is somewhat unique amongst studies of facial recognition, focusing on an applied version of line-ups used by some police jurisdictions. Future research would also benefit from conducting a comparative study between the methodology used in this study and the typical signal-detection methodology to see if the results generally align or if each methodology provides a unique insight into facial recognition.

### 4.3. Conclusions

The present study was undertaken to investigate the impact of consistency of handedness on the own-race bias in facial recognition. It was hypothesized that inconsistently handed individuals, with their increased access to right-hemisphere-based processes, would show a reduction in the own-rice bias. It was also hypothesized that this bias reduction would likely be the result of increased memory performance for inconsistently handed individuals. Results indicated that inconsistently handed individuals did show less own-race bias than did consistently handed individuals. This effect, however, seemed to be driven primarily by consistently handed individuals who showed an increase in performance for their own racial group but a reduction in performance for the outgroup (relative to inconsistently handed individuals). Although a clear understanding of the mechanisms by which the degree of handedness can influence the own-race bias cannot be determined from this study, there is ample evidence that handedness does play some role. Future research into the impact of handedness on the own-race bias should be fruitful in furthering our theoretical understanding of handedness as well as investigating practical implications of this stream of research.

## Figures and Tables

**Figure 1 brainsci-15-00828-f001:**
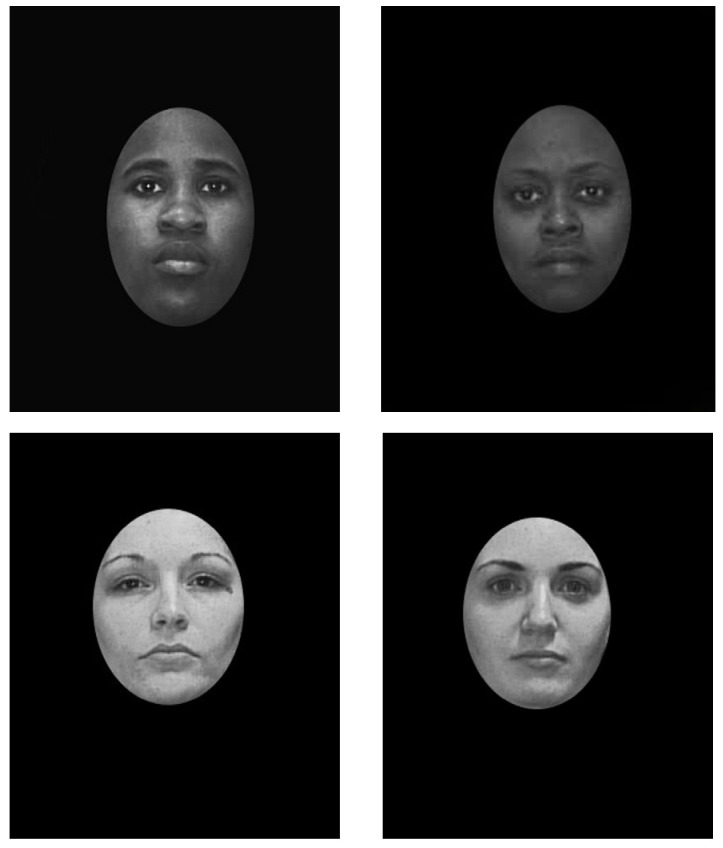
Example stimuli images.

**Figure 2 brainsci-15-00828-f002:**
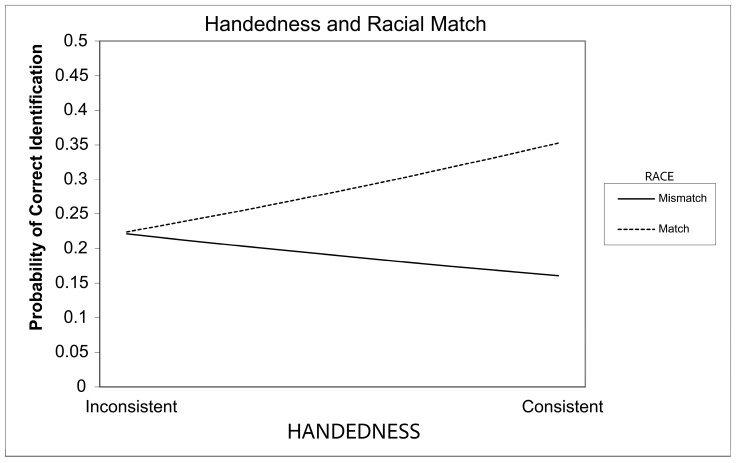
The impact of handedness and racial match, between the participant and the target face, on the probability of a correct facial identification. The *x*-axis represents moving from lower handedness scores (more inconsistently handed) toward higher handedness scores (more consistent handedness) continuously across a logarithmic scale.

**Table 1 brainsci-15-00828-t001:** Handedness frequencies.

EHI Score	Frequency	Percent
−100 to −75	10	4.65
−70 to −45	9	4.19
−40 to −15	5	2.33
−10 to −5	2	0.93
0	2	0.93
5 to 10	1	0.47
15 to 40	20	9.30
45 to 70	62	28.84
75 to 100	104	48.37
Total	215	

**Table 2 brainsci-15-00828-t002:** Logistic regression model parameter estimates.

Parameter	*b*	*SE*	*Wald*	*p*	OR
Constant	−1.457	0.175	69.08	<0.001	0.233
Racial Match	0.532	0.233	5.217	0.022	1.703
Handedness	−0.197	0.168	1.363	0.243	0.822
Race × Handedness	0.514	0.234	4.813	0.028	1.671

Note. Handedness ranged from 0 to 100 (inconsistent–consistent handedness). Correct answers were set as reference for the dependent variable. Match between participant and target race was set as the reference category in the predictor variable. Race in the interaction above refers to the racial match variable.

**Table 3 brainsci-15-00828-t003:** Descriptive answer types for the racial match variable.

Race	Hit	FI-After Miss	FI-Before Miss	Miss
Match	62	1	146	6
Mismatch	42	2	166	5

Note: Hit refers to a correct identification, FI-After Miss refers to a false identification, after the target face was already seen, FI-Before Miss refers to a false identification before the target face was seen, and Miss refers to an exhaustion of all facial stimuli without the participant selecting a face.

## Data Availability

The data supporting this study can be downloaded at http://doi.org/10.17632/bngzrxv7bk.2.

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
