# Peer review of "The Effects of Handedness Consistency on the Identification of Own- and Cross-Race Faces"

_brainsci, 2025, doi:10.3390/brainsci15080828_

Round 1
Reviewer 1 Report
Comments and Suggestions for Authors
I appreciate the opportunity to review the manuscriptentitled “The Effects of Consistency of Handedness on theIdentification of Own- and Cross-race Faces.”
The manuscript presents reasonable consistency; however, some methodological concerns remain andshould be addressed to improve the scientific rigor of thestudy. I outline these points below:
Abstract: The methodology and results sections in theabstract require further elaboration. Specifically, it isrecommended that the authors include more detailedinformation regarding eligibility criteria, group formationprocedures, group characteristics, and statistical analyses. In the results section, the inclusion of quantitative data isnecessary to substantiate the findings.
Introduction: The introduction demonstratestheoretical proficiency and conceptual clarity. However, certain aspects could be refined to enhance coherence, clarity, and scientific impact, thereby strengthening therationale of the study. For instance: (1) At various points, the text appears redundant, with different paragraphspresenting similar ideas in slightly different wording. A more concise and objective approach is advisable; (2) The section blends perceptual, cognitive, and social theorieswithout clearly delineating them. It would be helpful tobetter define transitions among experience-based theories, perceptual processing theories, and integrative models; (3) A more explicit explanation is needed regarding howinterhemispheric integration, linked to inconsistenthandedness, may influence categorization/individuationprocesses or holistic processing.
Materials and Methods: The data collection periodshould be specified. In the Materials subsection, technicaldetails about the image modifications (e.g., pixel dimensions, brightness, contrast, background color) shouldbe included. In the Participants subsection, please providemore detailed information regarding recruitmentprocedures and explicitly state the inclusion and exclusioncriteria. I also recommend organizing the paragraphstructure by topic (e.g., overall sample, racial distribution, handedness) to improve readability and comprehension.Furthermore, measures of central tendency and standard deviations should be reported in the Results section, notunder Methods.
Procedure: More details should be provided about thetesting environment (e.g., lighting conditions, presence ofexternal distractions, and mitigation strategies). Clarify theaverage duration of the task to aid interpretation of theresults. Additional information regarding randomizationprocedures between groups is also warranted.
Discussion and Limitations: The manuscript lacks a clear discussion of its limitations and the measures takento address them. I also suggest that the authors concludethe paper by summarizing the main findings in relation tothe study objectives and highlighting their implications for future research.
Author Response
Comments and Suggestions for Authors
I appreciate the opportunity to review the manuscript entitled “The Effects of Consistency of Handedness on the Identification of Own- and Cross-race Faces.”
The manuscript presents reasonable consistency; however, some methodological concerns remain and should be addressed to improve the scientific rigor of the study. I outline these points below:
Abstract: The methodology and results sections in the abstract require further elaboration. Specifically, it is recommended that the authors include more detailed information regarding eligibility criteria, group formation procedures, group characteristics, and statistical analyses. In the results section, the inclusion of quantitative data is necessary to substantiate the findings.
Response: Thank you for the comments and for reviewing the manuscript. We have expanded on the abstract to include additional details on the methodology and the overall statistical results as well as reorganizing the conclusion portion.
Introduction: The introduction demonstrates theoretical proficiency and conceptual clarity. However, certain aspects could be refined to enhance coherence, clarity, and scientific impact, thereby strengthening the rationale of the study. For instance: (1) At various points, the text appears redundant, with different paragraphs presenting similar ideas in slightly different wording. A more concise and objective approach is advisable; (2) The section blends perceptual, cognitive, and social theories without clearly delineating them. It would be helpful to better define transitions among experience-based theories, perceptual processing theories, and integrative models; (3) A more explicit explanation is needed regarding how interhemispheric integration, linked to inconsistent handedness, may influence categorization/individuation processes or holistic processing.
Response: Based on this comment, as well as comments from other reviewers, we have removed discussion of some of the presented theories that were more tangentially related to the hypotheses. Instead, we mention them only as other potential theories that readers should be aware of, but do not review them in detail. We also added in a citation that provides evidence that the right hemisphere is more adept at passing information across the corpus collosum to the left hemisphere, strengthening the argument that the right-hemisphere based processes would have a greater impact amongst inconsistently handed individuals with better interhemispheric interaction.
Materials and Methods: The data collection period should be specified. In the Materials subsection, technical details about the image modifications (e.g., pixel dimensions, brightness, contrast, background color) should be included. In the Participants subsection, please provide more detailed information regarding recruitment procedures and explicitly state the inclusion and exclusion criteria. I also recommend organizing the paragraph structure by topic (e.g., overall sample, racial distribution, handedness) to improve readability and comprehension. Furthermore, measures of central tendency and standard deviations should be reported in the Results section, not under Methods.
Response: We added in additional information regarding the composition of the photographs and added a figure containing example images from the study. We also included subsections for the paragraph as described in the comment.
Procedure: More details should be provided about the testing environment (e.g., lighting conditions, presence of external distractions, and mitigation strategies). Clarify the average duration of the task to aid interpretation of the results. Additional information regarding randomization procedures between groups is also warranted.
Response: We have added in additional details about the setting and procedure. We also clarified that the study took place in a single 30-minute research session. Unfortunately, because the sequential line-up ended once participants made a “yes” judgment, each participant experienced a study of different length. The time spent on the task can be calculated, but is likely not diagnostic of performance on the task beyond reflecting the timing of the “yes” judgment.
Discussion and Limitations: The manuscript lacks a clear discussion of its limitations and the measures taken to address them. I also suggest that the authors conclude the paper by summarizing the main findings in relation to the study objectives and highlighting their implications for future research.
Response: We expanded upon the discussion, limitations, and conclusion section of the manuscript.
Reviewer 2 Report
Comments and Suggestions for Authors
This is a manuscript describing results from a study that explores how consistency of handedness relates to the own-race bias (ORB) in facial recognition. The authors recruited 281 participants who completed a handedness inventory and then viewed two target faces (one same-race, one cross-race) followed by a distractor task and a sequential photo lineup in which they identified the previously seen faces. They found that inconsistently handed (ICH) individuals showed similar recognition accuracy for own- and cross-race faces, whereas consistently handed (CH) individuals were less accurate for cross-race faces, indicating that handedness may modulate the ORB in face recognition. The research question is novel, and the rationale is grounded in existing literature linking handedness to interhemispheric communication and episodic memory. The experimental design is clear, the analyses are appropriate, and the findings are potentially meaningful. That said, there are a few areas that need revision before the manuscript should be considered for publication.
Major Comments
The biggest issue I can identify is that the introduction is significantly longer than necessary and reads more like a literature review than a setup for a single empirical study. While much of the content is well written, it is overly detailed in areas that are only tangentially relevant to the hypothesis. The sections on face-space models, hemispheric specialization, and stereotype activation could be sharply condensed or removed. I recommend trimming this section substantially and focusing more tightly on material that directly motivates the experimental predictions.
In addition, the authors predicted that ICH individuals would show superior overall performance, based on prior work linking ICH status to episodic memory. However, the observed interaction was instead driven by poorer cross-race performance in CH individuals. This needs to be addressed a little more clearly in the Discussion. As written, the interpretation feels somewhat post hoc and speculative.
Further, the decision to split participants into CH and ICH groups based on a median split of the EHI score may be problematic and may reduce statistical power. Since the EHI is a continuous measure, it may be more appropriate to use it as a continuous predictor in regression models. At minimum, the authors should consider using supplementary analyses with EHI as a continuous variable.
Participants saw identical images at study and test phases, which raises concerns that participants may have relied on low-level image matching strategies rather than generalizable face recognition. Acknowledging this limitation is important, and future work would benefit from using varied images of the same individual across phases to better isolate face identity processing.
The “stop after yes” procedure limits the ability to fully evaluate signal detection properties such as false positives or bias, which also means sensitivity and specificity cannot be disentangled. The authors should acknowledge this limitation more explicitly.
Minor Comments
There are a few formatting issues throughout (e.g., misplaced punctuation in line 73, inconsistent reference formatting).
The figure legend for Fig 1 could be more descriptive. Currently, it lacks context for interpreting the plotted values.
Author Response
Comments and Suggestions for Authors
This is a manuscript describing results from a study that explores how consistency of handedness relates to the own-race bias (ORB) in facial recognition. The authors recruited 281 participants who completed a handedness inventory and then viewed two target faces (one same-race, one cross-race) followed by a distractor task and a sequential photo lineup in which they identified the previously seen faces. They found that inconsistently handed (ICH) individuals showed similar recognition accuracy for own- and cross-race faces, whereas consistently handed (CH) individuals were less accurate for cross-race faces, indicating that handedness may modulate the ORB in face recognition. The research question is novel, and the rationale is grounded in existing literature linking handedness to interhemispheric communication and episodic memory. The experimental design is clear, the analyses are appropriate, and the findings are potentially meaningful. That said, there are a few areas that need revision before the manuscript should be considered for publication.
Major Comments
The biggest issue I can identify is that the introduction is significantly longer than necessary and reads more like a literature review than a setup for a single empirical study. While much of the content is well written, it is overly detailed in areas that are only tangentially relevant to the hypothesis. The sections on face-space models, hemispheric specialization, and stereotype activation could be sharply condensed or removed. I recommend trimming this section substantially and focusing more tightly on material that directly motivates the experimental predictions.
Response: Thank you for the comments and for reviewing the manuscript. We have attempted to reduce the size of the introduction by removing, or reducing, several of the additional theories that were presented. We have, however, also added in additional research that was recommended by other reviewers.
In addition, the authors predicted that ICH individuals would show superior overall performance, based on prior work linking ICH status to episodic memory. However, the observed interaction was instead driven by poorer cross-race performance in CH individuals. This needs to be addressed a little more clearly in the Discussion. As written, the interpretation feels somewhat post hoc and speculative.
Response: We have attempted to rewrite the discussion to discuss this point more clearly and highlight it’s importance, especially for future research. The interpretations provided, however, are somewhat speculative as the hypothesized mechanism was not supported.
Further, the decision to split participants into CH and ICH groups based on a median split of the EHI score may be problematic and may reduce statistical power. Since the EHI is a continuous measure, it may be more appropriate to use it as a continuous predictor in regression models. At minimum, the authors should consider using supplementary analyses with EHI as a continuous variable.
Response: We apologize for the confusion. The median split was intended only as a descriptive exploration of the distribution of handedness scores. The actual logistic regression used a z-score standardized transformation of the absolute value score. We have added in additional language to the results to hopefully clarify this point.
Participants saw identical images at study and test phases, which raises concerns that participants may have relied on low-level image matching strategies rather than generalizable face recognition. Acknowledging this limitation is important, and future work would benefit from using varied images of the same individual across phases to better isolate face identity processing.
Response: We have added in additional language in the discussion , including highlighting the limitations as a separate subsection, to highlight this limitation and provide recommendations for future research.
The “stop after yes” procedure limits the ability to fully evaluate signal detection properties such as false positives or bias, which also means sensitivity and specificity cannot be disentangled. The authors should acknowledge this limitation more explicitly.
Response: We have attempted to highlight this stronger in the limitation section by separating it out into its own paragraph and beginning that paragraph by mentioning signal detection theory.
Minor Comments
There are a few formatting issues throughout (e.g., misplaced punctuation in line 73, inconsistent reference formatting).
Response: We have attempted to fix any minor formatting issues that we encountered, if you notice any additional issues, please let us know.
The figure legend for Fig 1 could be more descriptive. Currently, it lacks context for interpreting the plotted values.
Response: We have included additional labeling on the graph. We intentionally chose to leave off labeling of the handedness scale in the graph, however, because it is based on a logarithmic scale. Instead, we provided additional detail in the figure caption that the handedness variable goes from inconsistent handedness toward consistent handedness across a logarithmic scale.
Reviewer 3 Report
Comments and Suggestions for Authors
This paper hypothesized that handedness, a proxy for hemispheric specialization, would moderate own-race bias.
- It is suggested that the author enrich the content of the introduction and summarize the overall structure of the paper at the end of the introduction.
- The experimental part is not very convincing. It is suggested that the author conduct a comparative analysis with other methods.
- It is suggested that the author cite more papers published in the past three years.
- This paper lacks a conclusion. It is suggested that the author summarize the content of the article.
- Please consider discussing and analyzing the limitations of the proposed method.
Author Response
Comments and Suggestions for Authors
This paper hypothesized that handedness, a proxy for hemispheric specialization, would moderate own-race bias.
1.It is suggested that the author enrich the content of the introduction and summarize the overall structure of the paper at the end of the introduction.
Response: Thank you for reviewing the manuscript and providing comments. We have included additional research in the introduction section as well as removed some sections from the previous version that were only tangentially related to the research question in an attempt to tighten the introduction. We have also attempted to bolster the summary paragraph at the end of the introduction to properly reflect the research discussed and explain our hypotheses in line with this research.
2. The experimental part is not very convincing. It is suggested that the author conduct a comparative analysis with other methods.
Response: Unfortunately, this type of analysis would be impossible given the current data set. We have included this as a recommendation for future research in the discussion/limitation section.
3.It is suggested that the author cite more papers published in the past three years.
Response: We have attempted to add in additional research conducted in recent years. The study of consistency of handedness, however, represents a niche in the field and recent research that relates it to facial recognition is somewhat limited.
4.This paper lacks a conclusion. It is suggested that the author summarize the content of the article.
Response: We have separated the conclusion out as a subsection of the discussion and attempted to bolster this section.
5.Please consider discussing and analyzing the limitations of the proposed method.
Response: We have separated the limitations section out as a subsection of the discussion and attempted to bolster this section
Reviewer 4 Report
Comments and Suggestions for Authors
In my view, this is a good study. It looks at consistently handedness vs. inconsistently handedness as a proxy for lateral vs. bilateral processing of facial information.
On its own, that idea may be a bit of a stretch, but the suggestion is that inconsistently handed individuals engage in more holistic processing than consistently handed individuals is a proposition with which the data from the study appears to be consistent.
In addition to the general theoretical import of the case made by the authors, there are practical implications—e.g., regarding stereotypically biased processing and the ability to better recognize members of one's own ethnicity—are explored.
I don't have serious critiques of the study, but a few matters the authors may wish to elaborate on are as follows:
-
It may be helpful to define inconsistent handedness as the tendency to use one hand or the other for tasks that people normally perform with a single preferred hand. Incidentally, I am inconsistently handed. For example, I can play badminton, bat and use various tools with both hands, and I have above-average writing ability with both, though in the latter category I wouldn’t call myself ambidextrous. Specification in this direction might be helpful—e.g., does inconsistent handedness mean full ambidexterity, or are people with my capabilities included?
-
It might be helpful to lay out a little more of the brain science that allows for the supposition that inconsistent vs. consistent handedness implies more bilateral or lateralized processing—and why the former would lead to more or less in-group bias. Are other explanations for the association possible?
-
I am primarily familiar with literature discussing holistic processing of facial expressions, such as the following: Holistic Person Processing: Faces with Bodies Tell the Whole Story (2012); Body Cues, Not Facial Expressions, Discriminate Between Intense Positive and Negative Emotions (2012); Aesthetics and Action: Situations, Emotional Perception and the Kuleshov Effect (2021); Why Faces Don’t Always Tell the Truth About Feelings (2020)—
But the above literature in some ways is more holistic than the types of examples the authors are discussing, in that it explores how postural or environmental contexts affect facial processing (in this case of emotional facial expressions) in critical ways. The authors might consider including some discussion in this direction. For example, could part of the outcome they attribute to brain processing in inconsistently handed individuals also relate to a more bodily immersive embeddedness in the world? -
I'm a little surprised the authors did not include samples of the stimuli they used. This would help the reader. Also, can—at least in the studies mentioned above—be quite revealing in terms of appreciating the power (or lack thereof) of the propositions on offer.
The English is mostly fine but I always tick that box since I never encounter perfection
Author Response
Comments and Suggestions for Authors
In my view, this is a good study. It looks at consistently handedness vs. inconsistently handedness as a proxy for lateral vs. bilateral processing of facial information.
On its own, that idea may be a bit of a stretch, but the suggestion is that inconsistently handed individuals engage in more holistic processing than consistently handed individuals is a proposition with which the data from the study appears to be consistent.
In addition to the general theoretical import of the case made by the authors, there are practical implications—e.g., regarding stereotypically biased processing and the ability to better recognize members of one's own ethnicity—are explored.
I don't have serious critiques of the study, but a few matters the authors may wish to elaborate on are as follows:
1. It may be helpful to define inconsistent handedness as the tendency to use one hand or the other for tasks that people normally perform with a single preferred hand. Incidentally, I am inconsistently handed. For example, I can play badminton, bat and use various tools with both hands, and I have above-average writing ability with both, though in the latter category I wouldn’t call myself ambidextrous. Specification in this direction might be helpful—e.g., does inconsistent handedness mean full ambidexterity, or are people with my capabilities included?
Response: Thank you for reviewing the manuscript and providing comments. We have attempted to clarify that inconsistent handedness is a broader category that would include ambidextrous individuals but also includes individuals with a dominant handedness who do at least some activities with their non-dominant hand.
2. It might be helpful to lay out a little more of the brain science that allows for the supposition that inconsistent vs. consistent handedness implies more bilateral or lateralized processing—and why the former would lead to more or less in-group bias. Are other explanations for the association possible?
Response: We have included a citation that provides evidence that the right-hemisphere sends additional axonal projections across the corpus collosum to the left-hemisphere, and that those signals communicate quicker, than do signals coming from the left-hemisphere to the right. We also cited research showing that processes of faces can differ based on in-group/out-group status when the categorization is based on a highly polarized/affective situation.
3. I am primarily familiar with literature discussing holistic processing of facial expressions, such as the following: Holistic Person Processing: Faces with Bodies Tell the Whole Story (2012); Body Cues, Not Facial Expressions, Discriminate Between Intense Positive and Negative Emotions (2012); Aesthetics and Action: Situations, Emotional Perception and the Kuleshov Effect (2021); Why Faces Don’t Always Tell the Truth About Feelings (2020)—
But the above literature in some ways is more holistic than the types of examples the authors are discussing, in that it explores how postural or environmental contexts affect facial processing (in this case of emotional facial expressions) in critical ways. The authors might consider including some discussion in this direction. For example, could part of the outcome they attribute to brain processing in inconsistently handed individuals also relate to a more bodily immersive embeddedness in the world?
Response: Thank you for the recommendations. We have read the recommended articles and sighted several of them in the revised manuscript. We have also mentioned this idea of embeddedness as a potential mechanism related to inconsistent handers increased likelihood to be influenced by context.
4. I'm a little surprised the authors did not include samples of the stimuli they used. This would help the reader. Also, can—at least in the studies mentioned above—be quite revealing in terms of appreciating the power (or lack thereof) of the propositions on offer.
Response: We have included examples of the stimuli in the materials section.
Round 2
Reviewer 1 Report
Comments and Suggestions for Authors
Thank you for your thoughtful and careful edits.
Reviewer 3 Report
Comments and Suggestions for Authors
This paper can be accepted.